# Development of a Proposal for a Program to Promote Positive Mental Health Literacy among Adolescents: A Focus Group Study

**DOI:** 10.3390/ijerph20064898

**Published:** 2023-03-10

**Authors:** Joana Nobre, Helena Arco, Francisco Monteiro, Ana Paula Oliveira, Carme Ferré-Grau, Carlos Sequeira

**Affiliations:** 1Health School, Polytechnic Institute of Portalegre, 7300-555 Portalegre, Portugal; 2Faculty of Nursing, University of Rovira i Virgili, 43003 Tarragona, Spain; 3VALORIZA—Research Centre for Endogenous Resource Valorization, Polytechnic Institute of Portalegre, 7300-555 Portalegre, Portugal; 4Comprehensive Health Research Centre (CHRC), 7000-811 Évora, Portugal; 5Group Innovation & Development in Nursing (NursID), Centro de Investigação em Tecnologias e Serviços de Saúde (CINTESIS), 4200-450 Porto, Portugal; 6Nursing School of Porto, 4200-072 Porto, Portugal

**Keywords:** adolescents, mental health literacy, positive mental health, qualitative research

## Abstract

Over the last years, there have been several studies that have shown insufficient levels of adolescents’ mental health literacy (MHL). Knowledge about intervention programs that promote positive mental health literacy (PMeHL) among adolescents is still very scarce. In this sense, we defined as objectives to identify and describe the necessary components to design a program proposal that promotes adolescents’ PMeHL. We conducted an exploratory, descriptive, qualitative study using two focus groups in July and September 2022 with an intentional non-probability sample of eleven participants (nine professional experts and two adolescents). Data were analyzed using content analysis, using NVivo^®^ 12 software (version 12, QRS International: Daresbury Cheshire, UK). We obtained a total of four categories and eighteen subcategories: structure (context; format; contents; length and frequency; pedagogical methods; pedagogical techniques; resources; denomination), participants (target group; program facilitators), assessment (timing; evaluation instruments), other components (planning, articulation and adaptation; involvement; training; special situations; partnerships; referral). The perspectives of the professional experts and of the adolescents that we obtained from this study contributed to the design of a proposal for a program to promote adolescents’ PMeHL.

## 1. Introduction

The mental health of adolescents has been a subject of interest in recent years, not always for the best of reasons. The high prevalence of mental disorders is a reality [1,2], but above all, the insufficient level of mental health and mental health literacy among adolescents is a cause for concern [3,4,5].

We are aware that adolescence is a turbulent transitional period in the life of a human being, because it is full of marked and rapid changes in biological terms, which are also reflected in mental, psychological and emotional terms, and where brain neuroplasticity is high [2,6]. This is why adolescence is considered a privileged stage of human development for investment in the implementation of interventions to promote mental health literacy (MHL) but especially to promote PMeHL [2,7,8,9].

These concepts of health literacy (HL) and MHL have been gaining more expression and interest in the last decades, not only by health and education professionals, but also by researchers. According to the integrated model proposed by Sørensen et al. [10], HL refers to the “the knowledge, motivation and competencies of accessing, understanding, appraising and applying health-related information within the healthcare, disease prevention and health promotion setting, respectively” (p. 80), and is strongly based on the dimensions of access, understanding and application of health-related information and health services [11]. In turn, the concept of MHL was initially defined by Jorm et al. [12] and has been updated over the years by Jorm and by Kutcher and colleagues; MHL currently involves four dimensions: understanding how to achieve and maintain good mental health; understanding mental disorders and their treatments; decreasing the stigma related to mental disorders; and increasing the effectiveness of help seeking [13,14].

The concept of positive mental health (PMH) has also clearly emerged in recent years in the health field, due to its salutogenic dimension, and although it does not have a universal definition, it is related to an individual’s ability to understand themself and their environment in order to optimize daily functioning in relation to themself and others [15]. Associated with this concept is the Multifactor Model of Positive Mental Health developed by Lluch-Canut and consisting of six interrelated factors: personal satisfaction (Factor 1), pro-social attitude (Factor 2), self-control (Factor 3), autonomy (Factor 4), problem solving and self-actualization (Factor 5) and interpersonal relationship skills (Factor 6) [15].

Recently, Carvalho et al. [16] published a study on the conceptual analysis of PMeHL, in which they concluded that it is a dynamic concept, that it is one of the components of mental health literacy, and that it has the following attributes: competence in problem-solving and self-actualization; personal satisfaction; autonomy; relatedness and interpersonal relationship skills; self-control; and prosocial attitude [16]. Thus, the relationship between the concept of PMH and PMeHL is evident.

School has been recognized by several researchers as a privileged context for the promotion of HL and MHL in children and adolescents [2,7,9,17,18,19,20], for being the environment where adolescents spend more time [19,21], for being a place where adolescents come into contact with a huge diversity of people with diverse characteristics and ages [2,7], and where they are more available and more curious to develop knowledge and competencies [9,17].

Therefore, we must invest in interventions that promote adolescents’ PMeHL, so that they acquire competencies that allow them to deal with and experience all of the normal changes in this stage of human development in a healthy way, thus making a huge contribution to the future of having mentally healthy and resilient adults [2,6,22]. This is a wake-up call for health professionals and education professionals to implement interventions with adolescents, but also for researchers to build those interventions.

In this sense, and continuing our research to date in this area, we gathered a group of experts and sought to explore their perspectives on a PMeHL program for adolescents, using the focus group technique in order to obtain information to design an intervention proposal.

In order to identify and describe the components necessary to design a program proposal to promote adolescents’ PMeHL, we used the new framework for developing and evaluating complex interventions [23] from the UK Medical Research Council (MRC) in collaboration with the National Institute of Health Research (NIHR). This framework consists of four research phases that can be carried out in the context of complex interventions (development or identification of the intervention, feasibility, evaluation, and implementation) and contemplates core elements in each phase (considering context; developing and refining program theory; engaging stakeholders; identifying key uncertainties; refining the intervention; and economic considerations) [23]. In our study, we are in the intervention development phase, and we tried to take into special consideration the following *core elements*: (1) “*considering context*”, which in our case is the school context, and (2) “*engaging stakeholders*”, such that in the focus groups, besides health and education professionals, who are the potential facilitators of the intervention, we also included adolescents, who are the target group.

The research question that guided the present study was the following: What are the necessary components for designing a program to promote PMeHL among adolescents?

## 2. Materials and Methods

### 2.1. Study Design

A qualitative, descriptive and exploratory study was conducted using a focus group [24] and content analysis techniques [25]. The *Consolidated criteria for reporting qualitative research* (COREQ) checklist [25] was used as a guide for writing this paper.

### 2.2. Participant Selection

The participants in this study, in both focus groups (F1 and F2), were professional experts with experience in the field of positive mental health and adolescence, and were also included in the panel on adolescents in the early adolescent phase (10–14 years) as one of the *core elements* within the new framework for developing and evaluating complex interventions [23].

In this study, we used the non-probability sampling method to select the experts in the field of PMH, HL, MHL and adolescence for the sample constitution, i.e., an intentional sample was used.

One of the researchers (J.N.) sent an email to the professional experts inviting them to participate in the study, which contained the link to the online informed consent form and an explanation about the study. The professional experts were invited after discussion and consensus of the research team, based on their expertise, experience and work done. The selected professional experts met the mandatory inclusion criterion of wanting to participate voluntarily in the study, and at least two more of the following inclusion criteria: (a) having professional experience of at least 5 years; (b) being a health professional or a researcher or teacher in basic education; (c) having a master’s or doctoral degree; (d) having experience working with/researching adolescents; (e) being familiar with the concepts of PMH, HL and MHL.

Regarding the adolescents, the research team had access to their names and contacts by e-mail through the teachers of one of the schools in the Alentejo region of Portugal, who had previously contacted the adolescents and their parents/legal representatives, explained the objectives of the study and asked for their participation. An e-mail was sent to the adolescents’ parents/legal representatives by one of the researchers of the team (J.N.) with the formalization of the invitation to participate in the study and with the link to fill out the online informed consent form. The adolescents who were selected for participation in this study cumulatively met the following inclusion criteria: (a) being between 10 and 14 years of age; (b) being representative of their class at school; (c) not knowing any of the selected professional experts; (d) having informed consent authorized by their parents/legal representatives; (e) wanting to participate voluntarily in the study.

Given that the literature indicates that the panel of participants in a focus group can range from four to twelve members [24], and in order to ensure a sufficient number of experts in our sample, fifteen potential experts were invited (twelve professional experts and three adolescents), of which three declined (two professional experts and one adolescent), twelve accepted, and eleven actually participated in the study. Participants who refused or dropped out cited personal reasons as justification. Participation by experts and adolescents was voluntary, and there were no monetary compensations or other offers as incentives for participation.

### 2.3. Setting

Two focus groups were conducted by videoconference, through the Zoom platform, in order to facilitate the presence of participants, since they came from different regions of Portugal. In the first focus group, 11 participants were present, and in the second focus group 8 of the 11 participants were present. Following the methodological guidelines of Krueger and Casey [24], in addition to the participants, 2 members of the research team were also present, where one of the researchers played the role of moderator (J.N.), and the other researcher played the role of assistant moderator (H.A.).

### 2.4. Data Collection

The data collection for this study was performed through two focus groups, both directed to the same participants, using a semi-structured interview guide, in order to allow a greater degree of freedom in the participants’ answers, and the questions were constructed according to the objectives of the study: (1) What is the relevant content for a program to promote positive mental health literacy among adolescents in the stage of early adolescence (10–14 years) in a school setting? (2) What length (total and per session) and frequency should the program have? (3) What strategies do you think are relevant to use (methods, pedagogical techniques and teaching resources)? (4) What is the specific school context in which this program should be implemented? (5) What requirements must participants have to be targeted by the program? (6) What are the characteristics/requirements that facilitators must possess to implement the program? (7) How should the program assessment be done? (8) When should the program assessment be done? (9) What name do you suggest for the program? (10) Do you have any other suggestions that you think are pertinent? To assess sociodemographic characteristics, the researchers created an ad hoc form, through which it was possible to collect the following data: age, gender, academic qualifications, main professional activity, number of years of professional experience and number of years of experience working/researching with adolescents.

The first focus group was held in July 2022 and the second in September 2022, with a duration ranging from 90 to 120 min. During the focus groups, the moderator (J.N.) was responsible for presenting the objectives of the study and conducting the interviews, encouraging the intervention of all participants, especially the adolescents, in order to prevent them from feeling inhibited among the professionals; the assistant moderator (H.A.) was responsible for observing the focus group dynamics and taking supporting notes.

We used audio and video recordings of each focus group, duly authorized by the participants, and written supporting notes. Transcripts of the focus groups were made after the focus groups ended and were not returned to the participants for possible corrections or feedback.

After the second focus group, the authors considered that data saturation was achieved, as the response pattern of the participants was consistent and no new relevant information was obtained, which is consistent with what the literature says, i.e., that two to three focus groups are sufficient to capture about 80% of the main themes/categories [26].

### 2.5. Data Analysis

The verbatim transcription of each focus group was done by the first author (J.N.) and checked by the second author (H.A.). The data were then analyzed using content analysis according to Bardin [27] in its different phases: (1) pre-analysis, (2) exploration of the material and (3) treatment of results, inference and interpretation. In the pre-analysis phase, the transcripts and the assistant moderator’s notes were subjected to “*floating*” reading and editing procedures. In the material exploration phase, the first and second authors proceeded to the coding of the data and the researchers’ triangulation to minimize biases, and deductively coded four categories (structure; participants; assessment; others) and twelve subcategories (context, format, contents, length and frequency, pedagogical methods, pedagogical techniques, resources, denomination, target group, program facilitators, timings, evaluation instruments). Finally, in the results treatment phase, the name of the category “others” was changed to “other components” and six more subcategories emerged inductively (planning, articulation and adaptation, involvement, training, special situations, partnerships, referral).

The context units selected to illustrate the results obtained were identified by the code that was assigned to each participant in order to ensure anonymity, e.g., P1_SpNurs1_F1 means that the context unit comes from participant 1 who is a specialist nurse and participated in the first focus group.

Throughout the analysis procedure, the two authors (J.N. and H.A.) were concerned with the observation of the objectivity and pertinence of the categories, allowing this process to systematize them, in a reconfiguration procedure until reaching the final “Tree of Nodes”. We also observed validity, linking the objectives of the work, the emerging categories and the content included. We checked for exhaustiveness, ensuring the inclusion of input from a variety of data sources. Rigor was always a concern, in relation to the theme; the use of various informants and experts was a resource to ensure credibility; and transferability was observed, making rigorous reports in order to allow the transfer of knowledge supported by the results. We also emphasized the discussions between researchers, not only around the findings, but also on the methodological route, in an attempt to avoid distortions and once again control the reliability [28].

NVivo^®^ 12 software (version 12, QSR International, Ltd., Daresbury Cheshire, UK) was used to perform data analysis and treatment. The participants gave favorable feedback on their results after their analysis was returned to them.

## 3. Results

### 3.1. Characteristics of Participants

In focus group 1, eleven participants were present (P1–P11), of which nine were professionals (P1–P9) and two were adolescents (P10–P11). Focus group 2 was attended by eight of the eleven participants (P1–P2, P5 and P7–P11).

Of the total of eleven participants who took part in the focus groups, the majority were female (88.9%). The group of professional experts was composed of five specialist nurses in mental health nursing and psychiatry, one researcher, one psychologist, one child psychiatrist and one teacher in basic education; their ages ranged from 26 to 57 years; most of them had a master’s degree (66.7%); their professional experience ranged from 5 to 28 years; and their experience in research/work with adolescents ranged from 0 to 27 years. The adolescent group consisted of two adolescents, both 14 years old and attending the ninth grade. A more detailed view of the participants’ characteristics can be found in Appendix A.

### 3.2. Categories and Subcategories

With the two focus groups, we obtained in total four categories and eighteen subcategories, as shown in Figure 1. While performing phase 3 of the content analysis process, according to Bardin [27], i.e., the phase of treatment of results, inference and interpretation, the following six subcategories emerged inductively: planning, articulation and adaptation; involvement; training; special situations; partnerships; referral. All other subcategories had been deductively identified before the focus groups were conducted. A detailed view of the tree nodes showing all categories and subcategories as well as the number of references included in each can be found in Appendix A.

#### 3.2.1. Structure

***Context.*** The participants suggested that the implementation of the program should take place mainly in citizenship classes, as this is the subject whose contents are best suited to the theme of the program we are designing (P4_Teach_F1, P9_SpNurs5_F1, P10_Ad1_F1, P11_Ad2_F1, P2_Psy_F2, P11_Ad2_F2); however, they also suggested the possibility of this program covering other subjects in the curriculum plan (P4_Teach_F1, P6_ChildPsy_F1):

*We have some disciplines in schools that can collaborate a lot with this* [program], *such as citizenship*. (P4_Teach_F1)

*Should ideally be a transversal intervention, not in a particular discipline; it could be included in the content of several disciplines*. (P6_ChildPsy_F1)

***Format.*** It was considered by the participants that the program should have a modular format (P1_SpNurs1_F1, P7_SpNurs3_F1), *organized by sessions* (P7_SpNurs3_F2, P8_SpNurs4_F2), and they highlighted the importance of being based on the adolescents’ needs (P7_SpNurs3_F1, P1_SpNurs1_F2, P2_Psy_F2):

*It can be more modular*. (P1_SpNurs1_F1)

*I think that being organized by sessions is perfect; there has to be a logical following*. (P7_SpNurs3_F2)

*It doesn’t make sense to me to do a comprehensive promotion or prevention program for everyone, maybe more targeted and more individualized to the needs of each one*. (P7_SpNurs3_F1)

***Contents.*** There was consensus among the participants that the contents should be based on the factors of the Multifactor Model of Positive Mental Health, i.e., personal satisfaction, pro-social attitude, self-control, autonomy, problem-solving and self-actualization, and interpersonal relationship skills (P1_SpNurs1_F1, P2_Psy_F1, P4_Teach_F1, P8_SpNurs4_F1, P10_Ad1_F1), and should have as background the health literacy matrix, especially the dimension of ‘apply’ (P1_SpNurs1_F1, P2_Psy_F1, P3_SpNurs2_F1, P5_Res_F1, P8_SpNurs4_F1, P1_SpNurs1_F2, P5_Res_F2, P8_SpNurs4_F2):

*If we intend to have an intervention, a literacy program and to focus on positive mental health, then it is important that we base our content on the foundations of positive mental health*. (P8_SpNurs4_F1)

*The issue of ‘access’, ‘understand’ and ‘apply’ is important for literacy, and* [it should be] *very focused on ‘apply’*. (P1_SpNurs1_F1)

The participants also emphasized the importance of addressing emotions in order to clarify them for adolescents and to differentiate them from diagnoses of mental disorders (P1_SpNurs1_F1, P2_Psy_F1, P3_SpNurs2_F1, P4_Teach_F1, P5_Res_F1, P9_SpNurs5_F1, P11_Ad2_F1, P1_SpNurs1_F2). They recommended that special emphasis be placed on positive emotions and praise (P4_Teach_F1, P5_Res_F1, P6_ChildPsy_F1, P9_SpNurs5_F1), as well as on coping/resilience strategies (P2_Psy_F1, P6_ChildPsy_F1):

*There is a confusion between emotions and diagnoses*. (P3_SpNurs2_F1)

(…) *maybe we won’t talk about all of them [emotions],; we talk about the primary emotions eventually for the level of development they [adolescents] are at, [and] we talk about the most primary emotions*. (P1_SpNurs1_F2)

(…) *to praise strength,* (...) *working on positive emotions*. (P5_Res_F1)

Also in this subcategory, participants pointed out that the content should be based on the adolescents’ needs (P7_SpNurs3_F1, P2_Psy_F2):

*According to the needs of the target audience itself*. (P2_Psy_F2)

In addition, it was suggested by some of the participants that the Delphi technique should be used to validate the content of each module and session (P1_SpNurs1_F2, P8_SpNurs4_F2):

(…) *to validate this* [session content], *is to look at it thoroughly, maybe with a Delphi technique; I think it’s the simplest way to do it*. (P8_SpNurs4_F2)

***Length and frequency.*** It was suggested by participants that the program must have a well-defined start and end date for implementation (P7_SpNurs3_F1) and that it should have a regular frequency during implementation to ensure a certain continuity. Every two weeks or monthly implementation was suggested (P1_SpNurs1_F1, P2_Psy_F1, P4_Teach_F1, P5_Res_F1, P6_ChildPsy_F1, P8_SpNurs4_F1, P9_SpNurs5_F1, P10_Ad1_F1, P11_Ad2_F1), with concern that the frequency of implementation needs to be adapted to the school context (P7_SpNurs3_F2, P10_Ad1_F2):

*An intervention program at the literacy level has to be tight; it has to have a beginning and an end*. (P7_SpNurs3_F1)

*Frequency every 2 weeks or monthly*. (P9_SpNurs5_F1)

*In relation to frequency, I think it can be variable; it differs from school to school,* [and] *it depends on each school’s citizenship project*. (P7_SpNurs3_F2)

Regarding the length of each session, participants proposed 45 min, which corresponds to one class period (P7_SpNurs3_F1, P9_SpNurs5_F1, P10_Ad1_F2, P11_Ad2_F2), but pointed out that the length may vary depending on the content to be covered and the dynamics to be carried out, and that in some cases two class periods may be required, i.e., 90 min (P1_SpNurs1_F1, P7_SpNurs3_F1, P8_SpNurs4_F1, P10_Ad1_F1, P1_SpNurs1_F2, P2_Psy_F2):

*Maximum duration of 45 min*. (P9_SpNurs5_F1)

*I think that the duration is very relative, because it is very much associated with the content*. (P10_Ad1_F1)

(…) *the time I think has to be between 45 and 90 min; if we can negotiate, 90 min is much easier*. (P1_SpNurs1_F2)

***Pedagogical methods.*** Participants recommended giving priority to the use of active/dynamic pedagogical methods (P1_SpNurs1_F1, P3_SpNurs2_F1, P8_SpNurs4_F1, P9_SpNurs5_F1, P10_Ad1_F1, P10_Ad1_F2), with a concern to be appropriate to the goal of each session (P1_SpNurs1_F1, P8_SpNurs4_1, P7_SpNurs3_F2, P8_SpNurs4_F2). In addition, they suggested using methods/models that have already demonstrated effectiveness (P6_ChildPsy_F1):

(…) *in a more dynamic way and not exactly being closed in a room and listening to what someone tells us*. (P10_Ad1_F1)

*If we have people who still do not have a basic knowledge of what we are going to explain, maybe it is important to start with a more expository issue, then move on to the most active strategies, so we already have a higher level of knowledge that is likely required for the application*. (P8_SpNurs4_F1)

(…) *trying to look for positive models that have already demonstrated success and try to reproduce them*. (P6_ChildPsy_F1)

***Pedagogical techniques.*** Several pedagogical techniques were listed by participants that they most recommend to be used during program implementation, such as role-play/theater (P1_SpNurs1_F1, P3_SpNurs2_F1, P8_SpNurs4_F1, P9_SpNurs5_F1, P10_Ad1_F1, P1_SpNurs1_F2), games (P1_SpNurs1_F1, P3_SpNurs2_F1, P11_Ad2_F1, P11_Ad2_F2), group dynamics (P1_SpNurs1_F1, P4_Teach_F1, P7_SpNurs3_F1), discussion and reflection (P8_SpNurs4_F1, P9_SpNurs5_F1, P9_SpNurs5_F2), group work (P9_SpNurs5_F1, P11_Ad2_F1), online information searches (P9_SpNurs5_F1) and movies/videos (P1_SpNurs1_F2).

The importance of having a diversity of pedagogical techniques in each session and throughout the program was also underlined (P1_SpNurs1_F1, P4_Teach_F1, P8_SpNurs4_F1):

*I think that a 50-min class has to have 4 or 5 different activities to be dynamic and to* [encourage] *interaction between the students and the facilitators of the program*. (P4_Teach_F1)

***Resources.*** During the discussion, some participants suggested including technology as a resource to be used in the program implementation (P2_Psy_F1, P3_SpNurs2_F1, P5_Res_F1), because it is very suitable to the adolescents’ preferences, although they warned about a careful use of these technologies (P2_Psy_F1, P3_SpNurs2_F1, P8_SpNurs4_F1) due to the obstacles/problems they may imply, pointing out that there are also other resources that can be used during the program (P4_Teach_F1, P10_Ad1_F1):

*Adolescents like means of application; for example, any intervention must have digital means. They are the Z generation, they are already the Alpha generation, they were born in the digital environment, and this must be present*.(P5_Res_F1)

*Technology is good, and it’s appealing, but it’s also distracting; there has to be a balance between these two ideas of technology, yes, but with some caution because we can have a lot of obstacles later in our program*. (P8_SpNurs4_F1)

*Schools sometimes don’t have internet and that doesn’t make it easy; however* (…), *they* [internet and apps] *are not the only thing we can work with*. (P4_Teach_F1)

***Denomination.*** Although some suggestions for possible denomination of the program were made, participants suggested that it would be more interesting if the name of the program was chosen based on suggestions from the adolescents themselves, for example through a competition or voting (P7_SpNurs3_F1, P11_Ad2_F1), keeping in mind that the denomination needs to be short (P2_Psy_F2, P7_SpNurs3_F2):

[To determine] *the name* (…) *do a competition, for example, to choose, or a student vote*. (P11_Ad2_F1)

*The name,* (...) *it has to be something small to stay in memory, that is easier for diction*. (P7_SpNurs3_F2)

#### 3.2.2. Participants

***Target group.*** According to the participants, the program we are designing should start being implemented from the fifth grade and then continue in the following school years (P2_Psy_F1, P4_Teach_F1, P6_ChildPsy_F1, P11_Ad2_F1), i.e., start covering adolescents from 10 years old on. Furthermore, they proposed keeping classes together (P9_SpNurs5_F1, P11_Ad2_F1), dividing them into small groups during the sessions (P4_Teach_F1, P5_Res_F1, P10_Ad1_F1, P11_Ad2_F1). The following context units demonstrate the achieved findings:

*Mental health literacy has to be worked on as early as possible, it has to start in the 5th grade* (…); *however, it has to be worked on throughout life*. (P4_Teach_F1)

*I think it’s important to keep the class, because we know our colleagues better*. (P11_Ad2_F1)

(…) *groups have to be smaller*. (P5_Res_F1)

***Program facilitators.*** Participants considered that the program facilitator team should consist of health professionals and school professionals (P1_SpNurs1_F1, P3_SpNurs2_F1, P4_Teach_F1, P9_SpNurs5_F1, P10_Ad1_F1, P11_Ad2_F1, P1_SpNurs1_F2) to enrich the program:

(…) *always as a team: School Health, teachers, everyone, always a team*. (P1_SpNurs1_F1)

*I feel that we all have to be there because that’s the only way to bring even more benefits because our views and our visions of everyone are important,* (...) *I speak* (...) *of nursing, I also speak of the school psychologist, I speak of the professionals that we have available*. (P1_SpNurs1_F2)

Furthermore, two very important aspects were highlighted in the program design; on the one hand, the team should be composed of at least two facilitators (P3_SpNurs2_F1, P7_SpNurs3_F1, P9_SpNurs5_F1), and on the other hand, the facilitating team should be maintained until the end of the program implementation (P7_SpNurs3_F1), to guarantee continuity:

*We need at least two program facilitators to implement the intervention/session plus the class director, who is assisting, and a psychologist from the school*. (P3_SpNurs2_F1)

*The reference person who starts the program stays until the end* (…) *in order to give continuity*. (P7_SpNurs3_F1)

#### 3.2.3. Assessment

***Timing.*** Participants considered that evaluations should occur before the application of the program and at the end (P1_SpNurs1_F1, P2_Psy_F1, P3_SpNurs2_F1, P4_Teach_F1, P5_Res_F1, P7_SpNurs3_F1, P8_SpNurs4_F1, P9_SpNurs5_F1, P10_Ad1_F1), as well as during the follow-up (P4_Teach_F1, P6_ChildPsy_F1, P7_SpNurs3_F1, P9_SpNurs5_F1, P1_SpNurs1_F2). At these moments, the adolescents would be subject to self-evaluation by filling out instruments that will allow us to verify the existence or not of improvement in the results. The following context units illustrate the results achieved:

*It is important that we have a pre and post assessment here to see if there is effectively a gain*. (P8_SpNurs4_F1)

[Follow-up] *Always! At least after 3 months*. (P9_SpNurs5_F1)

Adolescent assessment was also recommended in each session (P1_SpNurs1_F1, P2_Psy_F1, P4_Teach_F1, P5_Res_F1, P8_SpNurs4_F1, P9_SpNurs5_F1, P11_Ad2_F1, P1_SpNurs1_F2, P2_Psy_F2, P5_Res_F2), through a hetero-evaluation performed by the program facilitators, to understand if the adolescents are able to apply the themes addressed in the sessions, and to get answers to the program process indicators:

*It is important to have an assessment of the process* (…) *in each session. It even gives us feedback on the improvement of the sessions because we can always improve the program*. (P1_SpNurs1_F1)

[Ask the adolescents] *“can you apply this information of positive psychology and positive health literacy to your everyday life, even just one thing? If you did, what did you apply it to?” This is what enriches the competences of these adolescents*. (P5_Res_F2)

***Evaluation instruments.*** Participants suggested the use of instruments that already exist and are validated for the assessment of positive mental health and mental health literacy (P1_SpNurs1_F1, P7_SpNurs3_F1, P8_SpNurs4_F1). However, they stated that in order to be able to make a more specific evaluation of the results regarding positive mental health literacy, the most correct thing to do would be to create an instrument to evaluate the results of the application of this program, as well as observation grids for the process evaluation in each session (P1_SpNurs1_F1, P2_Psy_F1, P3_SpNurs2_F1, P5_Res_F1, P8_SpNurs4_F1, P9_SpNurs5_F1, P2_Psy_F2, P7_SpNurs3_F2). The following context units demonstrate the results we obtained:

(…) *there is Claudia’s instrument, then there is Bjørsen’s own study*. (P1_SpNurs1_F1)

(…) *throughout the sessions we have observation grids*. (P1_SpNurs1_F1)

*To have a questionnaire that gives you an answer, a test that gives an answer to your program, you will probably have to build an instrument*. (P8_SpNurs4_F1)

In addition, they recommended that the instruments should allow for the assessment of acquired competencies and not just knowledge (P1_SpNurs1_F2, P2_Psy_F2, P5_Res_F2, P8_SpNurs4_F2):

*The instruments of evaluation cannot only be effectively instruments of knowledge; there must also be competencies here*. (P1_SpNurs1_F2)

(…) *a scale of* (…) *perceived competence or self-efficacy; it’s always easier for us to see a difference here*. (P8_SpNurs4_F2)

#### 3.2.4. Other Components

***Planning, articulation and adaptation.*** Participants mentioned that the program should focus on practical things (P3_SpNurs2_F1, P5_Res_F1, P2_Psy_F2, P7_SpNurs3_F2, P9_SpNurs5_F2), should be adapted to the adolescents’ and context’s needs (P1_SpNurs1_F1, P4_Teach_F1, P2_Psy_F2, P9_SpNurs5_F2), that facilitators should be careful with the language used (P5_Res_F1, P9_SpNurs5_F1) to make sure the message is delivered correctly, and emphasized that, above all, good planning and articulation with the school is required (P3_SpNurs2_F1, P4_Teach_F1, P9_SpNurs5_F1):

*It is better to focus on simple and practical things and bring awareness and give tools*. (P3_SpNurs2_F1)

*It is good to have this flexibility and this possibility of being able to adjust the program to the needs of the context, the needs of the adolescents and the capacity of the school*. (P1_SpNurs1_F1)

[The program] *must be planned at the beginning of the school year so that we are successful in the implementation*. (P3_SpNurs2_F1)

***Involvement.*** According to the participants, it is important that in the implementation of the program, the adolescents themselves who are going to be targeted by the program are involved (P3_SpNurs2_F1, P7_SpNurs3_F1, P1_SpNurs1_F2, P2_Psy_F2, P5_Res_F2), as well as the parents/legal representatives (P2_Psy_F1, P3_SpNurs2_F1, P10_Ad1_F1, P2_Psy_F2) and the school (P2_Psy_F1, P3_SpNurs2_F1, P2_Psy_F2):

*Adolescents should have a voice and should be listened to in their needs*. (P3_SpNurs2_F1)

*I think it shouldn’t be something addressed just to us, but also to parents*. (P10_Ad1_F1)

(…) *with the involvement of the school*. (P2_Psy_F1)

***Training.*** It was mentioned by the participants that it is essential to have prior training for all facilitators about the program and its implementation (P5_Res_F1, P7_SpNurs3_F2, P8_SpNurs4_F2):

*So, I think it was important for us to talk about this, that in order to be applied, it is necessary to provide training to those who are going to apply it, whether they are citizenship teachers, which seems fine to me, or whether they are Class Directors*. (P7_SpNurs3_F2)

***Special situations.*** The participants warned that there may be adolescents with special educational and health needs, but they should not be excluded from the application of the program (P1_SpNurs1_F1, P3_SpNurs2_F1, P4_Teach_F1, P8_SpNurs4_F1, P9_SpNurs5_F1). They also advised that special care should be taken with immigrant adolescents (P7_SpNurs3_F1) and with adolescents with mental health concerns (P2_Psy_F2). The following context units illustrate the findings obtained:

*I think that no one from the class should be removed because they have some of these criteria. I think that* (…) *someone who has a special need can continue to participate; it may not have the best result we expected, but it is important that he/she continue in the session anyway*. (P8_SpNurs4_F1)

*If we have foreign students, at least have care taken in the translation and in explaining the terms*. (P7_SpNurs3_F1)

*Adolescents who already have some level of suffering associated,* [should not be] *excluded, but perhaps the approach with these* [students] *will have to be different*. (P2_Psy_F1)

***Partnerships.*** It was mentioned by the participants that it would be interesting to establish partnerships, for example, with institutions of higher education, for data analysis (P3_SpNurs2_F1):

(…) *partnerships* (...) *to do the work of research data, because in clinical practice, I cannot do research and clinical practice*. (P3_SpNurs2_F1)

***Referral.*** Finally, participants recommended procedures that should be in place if facilitators identify adolescents who need specialized support (P1_SpNurs1_F2):

(…) *because when we have to refer a situation that we have identified, we will have to refer it “outside ourselves”*. (P1_SpNurs1_F2)

Through the visualization of the word cloud generated during the content analysis, illustrated in Figure 2, we see that the most common words verbalized by the participants in the focus groups are “think”, “adolescents”, “class”, “knowledge”, “school”, “session”, “apply”, “needs”, “important”, “literacy”, “program”, “health”, with respectively a frequency of 2.38%, 2.06%, 1.52%, 1.41%, 1.30%, 1.30%, 0.98%, 0.98%, 0.76%, 0.76%, 0.54%, 0.54%, which demonstrates the importance that participants attribute to the construction of a literacy-promoting program targeting adolescents, inserted in their class, that is applied in school, and that contains sessions in which adolescents think about the knowledge they are accessing and apply it in their daily lives to promote their mental health, according to their needs.

## 4. Discussion

This study explored the perspectives of a group composed of professional experts and adolescents on the design of a proposal for a program to promote positive mental health literacy among adolescents. As result, we were able to identify and describe the components necessary to design the proposal of the mentioned program, and these components were grouped into four categories: structure, participants, assessment and other components.

Regarding the category *structure*, the participants suggested that the program should be offered to adolescents at school, mostly in citizenship classes, since schools are considered by several authors to be a privileged context for the promotion of mental health, HL and MHL [2,7,9,17,18,19,20]. They suggested that it should have a modular format, with several sessions, every two weeks or every month, each session lasting between 45 and 90 min, using mainly active pedagogical methods, using pedagogical techniques such as role-play, games, group dynamics, discussion and reflection, among others, and eventually resorting to the use of technology, as long as it is very carefully selected so that it does not become an obstacle. These suggestions from the participants are consistent with the studies developed by other authors, namely Parnell et al. [29], Morgado et al. [9,30], Laranjeira & Querido [31], Choi et al. [32] and Costa et al. [33]. The contents were thoroughly discussed by all participants, and the consensus was that the Multifactor Model of Positive Mental Health factors should be addressed, that is, personal satisfaction, pro-social attitude, self-control, autonomy, problem solving and self-actualization, and interpersonal relationship skills. Special attention should be given to clarifying some emotions, emphasizing positive emotions, emphasizing the practice of praise and teaching coping/resilience strategies. On one hand, these proposals of the participants are in line with the new framework for developing and evaluating complex interventions [23] in the sense that there is a theoretical model at the basis of a complex intervention, which in our case will be the Multifactor Model of PMH and the HL matrix. On the other hand, they are in line with what the study by Carvalho et al. [16] indicated are the attributes of the concept of PMeHL. The participants recommended that the contents should be approached based on the HL matrix, that is, ensuring that adolescents develop access, comprehension and use competences regarding the contents that will be worked on [10,11], emphasizing that this will be a great added value of this program and will differentiate it from others that already exist. Indeed, in a scoping review conducted by our research team in 2021, we found that the several studies included referred to programs or interventions promoting adolescent MHL; however, only a few were directed to the salutogenic dimension of MHL and promoted mainly access to knowledge and not adolescent competences in the dimension of ‘*apply*’ [34]. It was also suggested that the validation of the content of each session should be performed through the Delphi technique, which is often used in health sciences to obtain consensus on a particular subject or intervention [35], which is exactly what we intend to do in a future stage. Regarding the denomination of the program, it was suggested that a vote or competition should be held with the adolescents, in order to involve them in the process of building the program itself.

In the category ‘*participants*’, it was consensual that the program should have as its target group adolescents from the fifth grade on, and it was advised that the class to which the adolescents belong should not be broken up, but the class should be divided into small groups during the sessions [9,32,33,36]. Regarding the facilitators of the program, it was recommended that a team should be formed with health professionals and education professionals, which should remain unchanged from the beginning until the end of the program implementation, and that at least two facilitators must work together to run each session, which is in line with what is advocated by several authors [7,8,18,36].

In relation to the category ‘*assessment*’, participants advocated for evaluation before and after program implementation and also during the follow-up [9,37,38], where adolescents should be given instruments to evaluate PMeHL, either through instruments that already exist and are properly validated and adapted or through an instrument that the research team constructs specifically for program evaluation. In addition, they recommend that during each session, the adolescents should be evaluated by the program facilitators through the application of observation grids that the research team needs to create in order to obtain information not only about the adolescents’ progress, but also suggestions for improvement of the program in progress, as recommended by Richards & Halberg [37] and by Morgado et al. [9,30].

Finally, in the category ‘*other components*’, there are several extra suggestions that may contribute to a greater effectiveness and success of the program, which emerged from the participants’ responses throughout the focus group. In particular: (1) the need for good coordination with the school in terms of short- and medium-term planning; (2) the importance of involving the adolescents themselves in the planning and implementation of the program, giving priority to their expressed needs, their opinions and active participation in decision-making, as well as the involvement of the school and parents/legal representatives; (3) the requirement for the facilitating team to undergo training prior to the implementation of the program; (4) the need to adopt inclusive and more protective measures in case there are adolescents with special health and educational needs, or non-native speaking adolescents or adolescents with mental health concerns; (5) the extreme importance of having a previously defined referral circuit for specialized care for the adolescents that the facilitators identify during the implementation of the program; and (6) the establishment of partnerships with some entities, for example, for the analysis of the data resulting from the evaluation instruments, but which may also be important in the context of the aforementioned specialized care for adolescents. These suggestions are consistent with several published works [2,7,9,33,36].

It is noteworthy that in several subcategories (throughout the two focus groups) the participants emphasized that it was extremely important that the program be adapted to the adolescents’ needs and also to the needs and characteristics of the school context, which is in line with what the new framework for developing and evaluating complex interventions [23] advocates, that is, the flexibility that interventions must have in order to be effective.

Despite the methodological rigor and the relevance of the achieved findings, this study has some limitations. First, the results cannot be generalized, since we used a non-probabilistic sample. Second, the small number of adolescents that participated in the focus groups and their limited diversity in terms of characteristics may have limited the diversity of the perspectives obtained. Third, the unbalanced group of experts may have conditioned the representativity of the various professional groups. Lastly, both focus groups included the same participants.

## 5. Conclusions

In conclusion, the participants of this study expressed their ideas about the components that a program to promote adolescents’ PMeHL should contain, thus contributing to the design of the program proposal. They considered that it should be a program implemented at school by a team of facilitators composed of health and education professionals. This program should be aimed at adolescents from the fifth grade on, with a focus on citizenship classes, composed of several modules, with each module organized into sessions, each 45–90 min long, with a quarterly or monthly application frequency. Active teaching methods and techniques must be used. The contents should be based on the factors of the Multifactor Model of PMH and based on the dimensions of access, understanding and application of HL. This program should also include the application of adolescent outcome assessment instruments before and after implementation and at follow-up, as well as adolescent process assessment instruments at each session. To increase the program success, participants recommended: good coordination with schools; the involvement of adolescents, parents/legal representatives and the schools themselves; the adoption of inclusive and protective measures for the most vulnerable adolescents, which should include a referral circuit for specialized care; and the establishment of partnerships with other entities.

In terms of implications for clinical practice, we believe that in the future, this program will make an outstanding contribution to the promotion of adolescents’ mental health and well-being by supporting professionals in clinical practice (nurses, clinical psychologists, physicians) and educational professionals (teachers, educational psychologists) in their care of this group of the educational community.

It will be necessary to develop further research studies to obtain a program properly validated and adaptable to adolescents and their context, while having a standard base structure. This means that it is important to conduct Delphi studies for the validation of the contents of each session, to perform a pilot study, and after that to develop experimental or quasi-experimental studies.

In future studies, the participants in the focus groups should be more balanced, regarding their number and their characteristics of experts and adolescents, to provide a higher strength of the findings. In addition, separate focus groups should also be organized, i.e., one with experts and then one with adolescents, to enhance the expressivity of adolescents’ perspectives and to contribute to their help to researchers in the applicability of the proposals made by the experts. If possible, it would also be interesting to conduct focus groups with different groups of experts and with different groups of adolescents, to ensure an even higher degree of data saturation.

## Figures and Tables

**Figure 1 ijerph-20-04898-f001:**
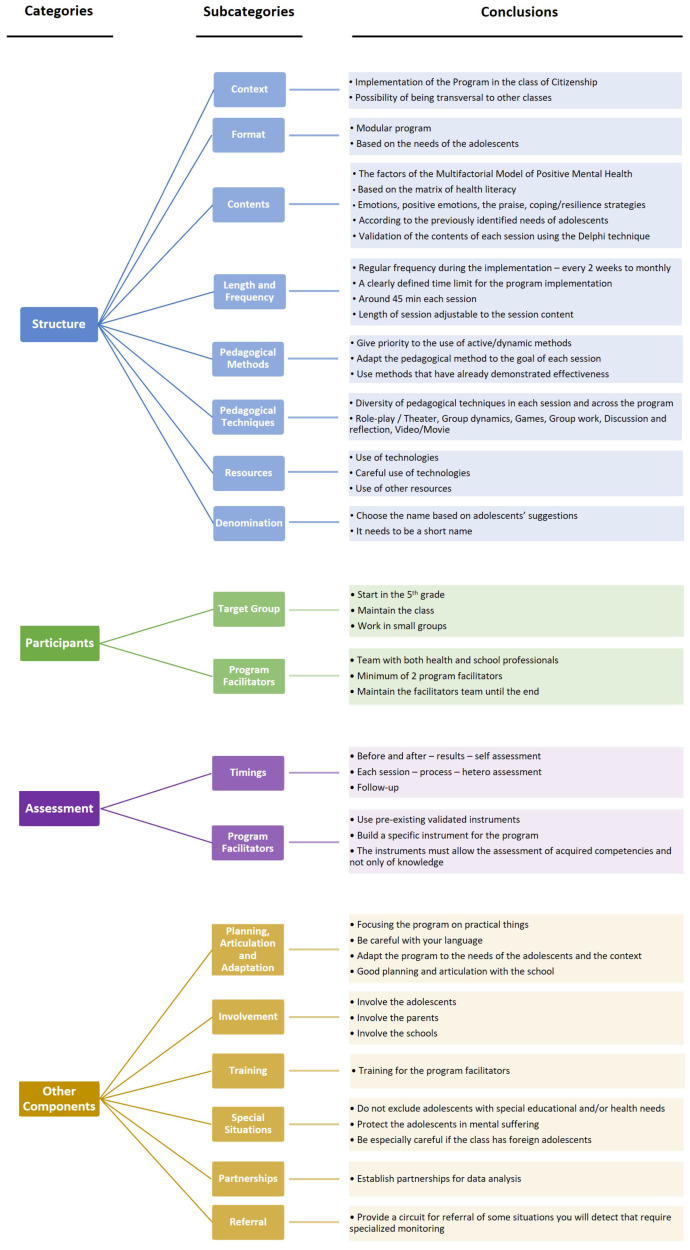
Categories and sub-categories resulting from the content analysis.

**Figure 2 ijerph-20-04898-f002:**
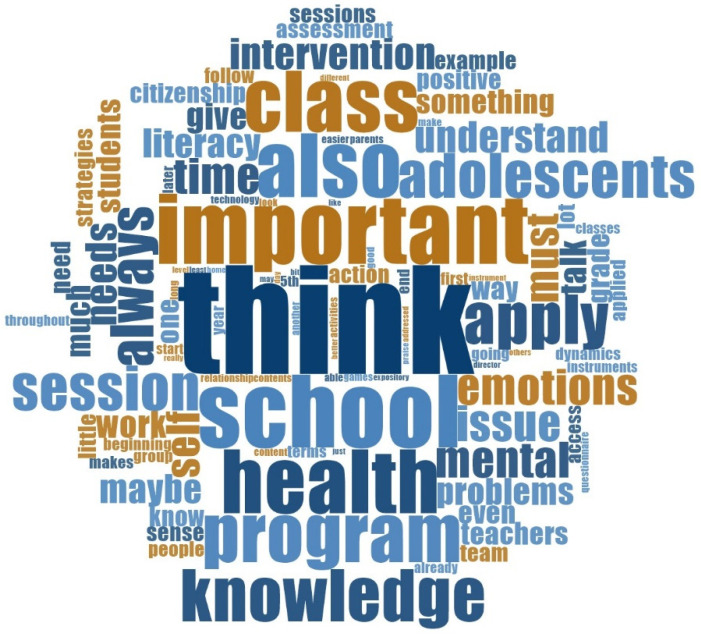
Word cloud of the most cited words by the participants (generated by the software NVivo^®^ 12).

## Data Availability

Data available on request due to ethical restrictions.

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
