# Peer review of "Development of a Proposal for a Program to Promote Positive Mental Health Literacy among Adolescents: A Focus Group Study"

_ijerph, 2023, doi:10.3390/ijerph20064898_

Round 1

Reviewer 1 Report

I have revised the article entitled Development of a Proposal for a Program to Promote Positive Mental Health Literacy among Adolescents: A focus group study. I found the study valuable for its practical implications for intervention, and for addressing a relevant subject, particularly in a post-pandemic moment when mental health has been challenged and requires particular attention. 

Despite this, there are some relevant methodological concerns, which the authors should justify and potentially include as limitations to their study and, hopefully, improve in future research efforts. Although I have commented on such issues in the PDF in the annex to this comment, I will summarize those concerns here:

a) the inclusion of only two adolescents who seem quite similar to one another (both classroom representatives, 14 years old and from the 9th grade of the same school). Wouldn't it be valuable to include more adolescent participants and express a wider range of future program participants - ages, genders, etc.?

b) including the adolescent participants in the same focus groups as the experts. It seems more sensible to organize at least 2 focus groups: one with the experts and one with the adolescents. The sequence I would propose would be experts - followed by adolescents. Adolescents could then face some of the proposals and help researchers decide on their applicability. By joining all participants, the adolescents' voices will most likely be less expressive than the voices of the experts, as is clear in the presentation of data. This has ethical as well as research implications.

c) having so many nurses and so few representatives of other areas of knowledge, including education. I have noticed that even when the theme was related to pedagogical intervention, nurses' voices were the most prevalent. 

d) having two focus group meetings with the same participants. It is impossible to speak of saturation when both groups' participants are the same. Only if more groups were constituted and were coming to the same conclusions would that be adequate.

Apart from these relevant concerns with methodology, I have appreciated the article's structure, figures, and overall organization. I have signaled some issues with English language translation in the document.

I wish the authors success with the continuing of this project.

Author Response

Dear Reviewer,

Please find attached in the PDF file our answers to your comments.

With our best regards,
Joana Nobre
On behalf of all the authors

Reviewer 2 Report

The study is interesting and adresses a relevant topic that focus on identify some of the main components for promoting adolescent positive mental health programs. One of the strenghts of this study the utilization not only of experts of differents fields of research and proffesional experience related with these programs as experts but of using adolescents as participants. As well as the use of a qualitative methodology for the study. 

Altought, I think there some things that can be improved to contribute make a better paper. Here are some of my comments.

For example, in the characteristics of participants section, it would be good to include a table with a brief description of the experts participating in the focus groups (e.g., gender, profession, years of experience, knowledge of positive mental health programs, HL, MHL, previous with adolescents, etc.). This table could show the different inclusion criteria that were met by the experts and adolescents and would contribute to provide clearer information about the sample and the specific characteristics of the participants in this study. It also contribute to make more clear the reading of the results when referring to the different voices of participants in the text with respect their comments on every dimensions (categories and subcategories) analysed of the components for the program. 

In future studies, the number of participants in the focus groups should be more balanced, including more students as participants, because having only two adolescents as participants in the focus groups vs 9 experts is not balanced.  

I found very interesting the qualiative analysis carried out and the categories and subcategories resulting from the content anaylisis. But I think authors should work more on the last category or dimension found named as "other components". Maybe this category should be divided in two, one could be named as "participation and inclusion", given that some of the subcategories are related with taking into account the inclusion and participation of students, families, schools in a more broader way. And perhaps the second subcategory should be named not as "other components", but as something different in a more specific and descripive way. For example, something as "planning and training".... But these names are only inferred from categories found in the description of the categories in the study. Of course this is only a suggestion to think on more descriptive and specific subcategories instead of using such a broad category such as "other components". 

Author Response

(The authors gave the same response as above.)

Reviewer 3 Report

Dear Authors:

Congratulations on a wonderful study and a well-crafted journal article. I only have some text editing comments and additions to the limitations of the study.

Line 49 has a grammar error.

Lines 92-96 is a run-on sentence. Please break this up into 2 or 3 sentences to make more sense to the reader.

On line 110 do you mean "non-probability sampling?"

Lines 474-477, this sentence does not make sense in English. Do you mean, "They also advised that special care should be taken with adolescents with mental health concerns and immigrant adolescents." 

Lines 492-494, this sentence does not make sense in English. Do you mean, "Finally, participants commented on the procedures that should be in place if facilitators identify adolescents who need specialized support." 

It seems that section 3.2.4 (line 443) needs to have subheadings to make it more organized. 

Lines 520-523, consider making this change: "...should be offered to adolescents at school, mostly in citizenship classes, since schools can easily promote mental health, Hl, and MHL." 

Lines 534-536, consider making this change: "Special attention should be given to clarifying some emotions, emphasizing positive emotions, offering praise, and teaching coping/resilllience strategies." 

Lines 536-541 is a run-on sentence. Please break this up into 2 or 3 sentences to make more sense to the reader.

Line 549 the word "apply." If this is in reference to a stage or a step in a model, it should be in italics.

Lines 520-553, this paragraph must be split. It seems the first split should occur at line 530. It seems the second split should occur at line 541. 

Line 565 "the" should be removed.

Line 568 "that at each" should be replaced with "that during each"

Lines 575-589 is a run-on sentence. Please break this up into 2 or 3 sentences to make more sense to the reader.

Lines 582-583 should be replaced with "...needs, non-native speaking adolescents, adolescents with mental health concerns..."

Lines 597-599. There are many more limits to this study.

Lines 603-610 is a run-on sentence. Please break this up into 2 or 3 sentences to make more sense to the reader.

Lines 610-612. This is not a sentence.

Lines 612-616 is a run-on sentence. Please break this up into 2 or 3 sentences to make more sense to the reader. There are also many grammar issues in this area.

Lines 622-625 is a run-on sentence. Please break this up into 2 or 3 sentences to make more sense to the reader. There are also many grammar issues in this area.

Author Response

(The authors gave the same response as above.)

Round 2

Reviewer 1 Report

I recommend that the authors at least include the suggestions they received and state they will consider for future studies, in the recommendations for future studies and limitations of the study sections of their discussion or conclusion since there are several methodological limitations. It is not enough to acknowledge these aspects privately in the comments to the reviewer - they should be reflected in the manuscript itself.

Author Response

Dear Reviewer #1,

Please find attached our response to your comments.

Kind regards,

Joana Nobre
